# Determinants of care-seeking practice for neonatal illnesses in rural Bangladesh: A community-based cross-sectional study

U. Tin Nu, Jesmin Pervin₀*, A. M. Q. Rahman, Monjur Rahman₀, Anisur Rahman

International Centre for Diarrhoeal Disease Research, Bangladesh (icddr,b), Dhaka, Bangladesh

* jpervin@icddrb.org

**Data Availability Statement:** All relevant data are within the manuscript and its Supporting Information files.

## Abstract

### Background

Proper utilization of skilled care services in neonatal illnesses is crucial to reduce neonatal morbidity and mortality. The study aimed to evaluate the level and factors associated with seeking care from skilled healthcare service providers for reported neonatal illnesses in rural Matlab, Bangladesh.

### Methods

This community based cross-sectional study was based on data from a randomly selected sample comprised of 2223 women who delivered live-born babies in 2014. Data were collected from June to October 2015 through a structured questionnaire. We used a multivariable logistic regression model and presented the results by adjusted odds ratios (AOR) with 95% confidence intervals (CI).

### Results

Of the neonates, 1361 (61.2%) suffered from at least one complication, and among these, 479 (35.2%) sought care from skilled healthcare service providers. In the multivariable logistic regression analysis, the participants' husbands' educational level, number of antenatal care visits, and place of childbirth were significantly associated with seeking skilled care for reported neonatal illnesses. The care-seeking from skilled healthcare service providers for neonatal illness was more than two times higher (AOR = 2.26, 95% CI = 1.51–3.39) in the group in which the participants' husband had attended school for more than 10 years as compared to the group in which they had attended school for less than six years. The AORs of seeking skilled care were 1.93 (95% CI = 1.42–2.62) and 2.26 (95% CI = 1.51–3.39) with the mothers receiving two to three and four or more antenatal care services, respectively, compared to the mothers with no or one antenatal care visit. Women who gave birth at a health facility were three times (AOR = 3.24, 95% CI = 2.50–4.19) more likely to seek skilled care for sick neonates compared to those who gave birth at home.

**Funding:** This work was supported by the USAID through Partnerships for Enhanced Engagement in Research (PEER) Health, Grant Number: 01122 (https://www.usaid.gov/bd; http://sites. nationalacademies.org/pga/peer/index.htm). The funder had no role in study design, data collection and analysis, decision to publish, or preparation of the manuscript.

**Competing interests:** The authors have declared that no competing interests exists.

## Conclusion

The utilization of skilled care for neonatal sicknesses was low in this rural setting in Bangladesh. The participants' husbands' higher school attendance, increased number of ANC visits, and facility delivery were positively associated with care-seeking from skilled healthcare providers for neonatal illness. The husbands with low school attendance should be targeted for intervention, and continue efforts to increase ANC coverage and facility delivery to improve neonatal health in this country's rural area.

## Introduction

Care during the neonatal period, the first 28 days of life, is crucial for survival as during the transition from fetal to a complex extra-uterine environment, the newborns are most vulnerable, and their developing organs are not mature enough. Globally, in the last two decades, we observed substantial improvement in the reduction of mortality of children under five years. However, this reduction is not evenly spread across all age groups. While the decrease in mortality in the age groups of one to 11 months and one to four years were 51% and 60%, respectively, the neonatal mortality reduced by only 41%. The global average of neonatal deaths contributing to the under-five mortality was estimated to be 47%, with 2.5 million neonatal deaths in 2017 [1]. Almost 90% of these deaths take place in low- and middle- income countries (LMICs) [2, 3]. In Bangladesh, the neonatal mortality rate is 30/1,000 live births, and it accounts for 67% of under-five child mortality [4]. Under the Sustainable Development Goal 3, the agenda is to end all preventable deaths of newborns and children under five years by 2030. All countries, including Bangladesh, aim to reduce neonatal mortality to less than 12/ 1,000 live births [5].

Globally, about 32% of women give birth at home [6]. Although the proportion of home-based delivery has substantially reduced in Bangladesh, still 50% of deliveries take place at home with no skilled birth attendants [4]. When a caregiver notices a sign of any neonatal illness at the household level, immediate consultation with a skilled healthcare provider is recommended to receive appropriate treatment. However, the available studies from countries in South Asia and Sub-Saharan Africa have reported that skilled healthcare for sick neonates was low [3, 7–9]. The use of skilled healthcare reportedly varies from 17% and 56% in Bangladesh [2, 10–13]. A significant proportion of caregivers resorted to unskilled healthcare providers such as traditional healers, untrained village doctors, and homeopaths [2, 10, 12, 14].

The utilization of a skilled healthcare provider is challenging and depends on individual and social factors. These include women's age [15], the women and their husbands' educational level [2, 7, 16], economic condition [15, 17], utilization of antenatal care, facility delivery, and postnatal care [2, 7–10, 18]. Studies also reported prompt identification of danger signs and perceived quality of care for using skilled healthcare providers as determinants [19, 20]. Furthermore, social factors such as autonomy of decision-making and accessibility to health facilities may also influence seeking skilled-care for sick neonates [18, 20]. While a strengthened health system, with essential services and human resources, is needed to address the problem, recognizing illness and caregivers' decision to reach the skilled healthcare providers are critical to reducing mortality.

To our knowledge, significant work has been done to implement maternal and child health programs, but neonates are still suffering from morbidities and mortalities related to illnesses

[13, 20, 21]. This suffering is partly attributed to the care-seeking practices of women for sick neonates. The systematic reviews have reported the low utilization of skilled healthcare services and emphasized the need for more studies on the care-seeking pattern from low-income countries [3, 20]. There is also a scarcity of population-based studies. Few studies evaluating the levels and factors related to skilled healthcare for neonatal illnesses in Bangladesh were either conducted a long time back or used data from post-intervention surveys [2, 10, 11, 13]. Therefore they do not represent the current situation of newborn healthcare-seeking pattern in the country. Further, to design targeted strategies and address the challenge contextually, it is essential to understand the pattern of care-seeking and the determinants of the utilization of skilled care services for sick neonates. Therefore, the present paper analyzed the baseline survey conducted as part of an intervention study to explore the pattern of care-seeking and the associated factors that influence mother's healthcare-seeking practices from skilled providers for reported neonatal illnesses in rural Matlab, Bangladesh.

## Materials and methods

### Study site, design, and participants

We conducted this cross-sectional study in two sub-districts–Matlab North and Matlab South in the district of Chandpur, Bangladesh. It is a low-lying deltaic plain, situated around 85 km southeast of Dhaka, the capital of Bangladesh. Around 90% of inhabitants are Muslims, and the remaining are mostly Hindus.

Bangladesh has a reasonably good health infrastructure. The delivery of primary healthcare services consists of three tiers:- Community Clinics (CC) at the village level, Union Health and Family Welfare Centers (UHFWC) at the union level, and Upazila Health Complexes (UHC) at the sub-district level. Generally, UHC serves as a primary health care referral facility with the emergency, inpatient, and outpatient services where doctors, nurses/midwives, and sub-assistant medical officers (SACMO) are the leading service providers. However, UHFWC and CC are operated by family welfare visitors (FWV) and community health care providers (CHCP), respectively. Also, community domiciliary health workers such as family welfare assistants (FWA) and health assistants (HA) provide health education on maternal and child health through routine household visitations [22–24].

This population-based cross-sectional study used the baseline survey data collected before implementing the Maternal, Neonatal, and Child Health Extension (MNCH-Ext) project. This project aimed to implement a successful model of a package of evidence-based intervention during pregnancy and delivery into the public health system [21]. In the MNCH-Ext project, we included women who gave birth to either a live-born or stillbirth or aborted her fetus between January and December 2014. These women were identified through household visits from February to April 2015. In each village, a landmark was selected to identify the first household, and then the houses were visited in a clockwise direction. About 7,018 women were listed through household visits. Among them, 6,741 women delivered live or stillborn babies, and 277 women had abortions. We used this sampling frame to select the required sample size to determine the prevalence of selected health indicators, including antenatal care visits, facility delivery, and knowledge, attitude, and practices of maternal and neonatal illnesses in the MNCH-Ext project area. This survey aimed to assess the comparability by selected health indicators between two study arms at baseline. With a precision of 3%, 95% confidence intervals, prevalence rate of 50% of health indicators, and 15% non-response rate, the required sample size was 1242 for each study arm. Out of 6741 deliveries available, 2,484 women were randomly selected using the SQL (structured query language) database by a researcher not engaged in the MNCH-Ext study. Out of randomly selected women, 2262

(89.5%) women gave consent and participated in the study. After excluding the women who delivered stillborn babies, 2223 women who delivered live-born babies were included in the present paper (Fig 1).

## Data collection

A structured questionnaire was used to collect data from the study participants. The interview questionnaire was prepared in English and translated to the local language, Bangla. A total of 20 trained female data collectors, fluent in the local language, and familiar with local norms, collected data from June to October 2015 by conducting face to face interviews at the participant's home. The questionnaire included questions on the following information:

**Neonatal danger signs.** Based on the recognized danger signs of neonatal illnesses by the World Health Organization (WHO) [25] that could endanger a newborn's life, seven neonatal signs were listed in the questionnaire. The danger signs were: i) fever, ii) difficulty in breathing, iii) jaundice, iv) diarrhea, v) umbilical cord infection, vi) convulsion, and vii) feeding problem. Women were asked to list any of these danger signs that their neonates had experienced within the first month of their life.

**Healthcare seeking practices.** Information on the type of care-seeking for the reported neonatal illnesses were collected and further categorized into skilled and unskilled care. If the care was utilized from a legally authorized service provider such as a doctor or a nurse or a midwife or a paramedic working either at government or private health facilities, it was considered skilled care. If the care was sought from a provider who was not legally authorized, such as a village doctor, a homeopath, a spiritual healer, a *kabiraj*, a traditional birth attendant (TBA) or a family member, it was considered as unskilled care.

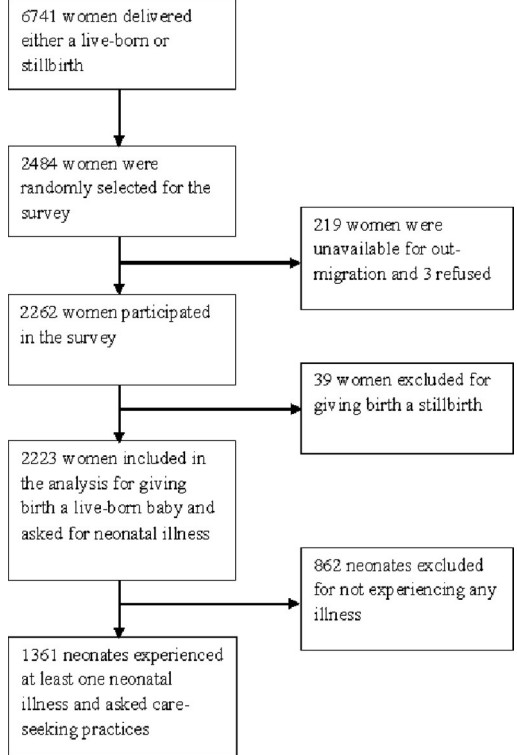

**Fig 1. Study profile.**

**Covariates.** We extracted information about the participant's age, education levels of the women and their husbands, socioeconomic status, number of ANC visits, and place of delivery. The educational status was determined by the year of attendance in schools and categorized into <6 years, 6–10 years, and >10 years of school attendance. Using principal components analysis, we created wealth scores based on household ownership of selected assets such as several consumer items (television, radio, watch, and furniture), dwelling characteristics (wall and roof material), drinking water, and sanitation facilities. The computed scores were further grouped into quintiles, wherein one was defined as the poorest and five as the richest [26]. The number of antenatal care visits was defined as the number of times care was received during pregnancy, and it was grouped into 0 or 1, 2–3, and ≥4 ANC visits. Information on place of the childbirth was also recorded and labeled as either home or facility delivery.

## Data analysis

The data were analyzed using both descriptive and analytical approaches. The study participants' characteristics were presented by proportions, mean (standard deviation, SD), or median. In the univariate logistic regression model, we assessed the association between the individual covariate and the outcome of interest. After that, all factors associated with the outcome were included in the multivariable logistic regression analysis. A backward stepwise approach was used to identify those independently associated with the outcome of interest ($P$-value < 0.05) to keep in the final model. In the final model, the factors were adjusted to each other. The logistic regression results were expressed as the odds ratio (OR) with a 95% confidence interval (CI). The data were analyzed by IBM SPSS Statistics, version 20 (IBM corporation, NY, United States).

## Ethical considerations

Ethical approval was obtained from the Research and Ethical Review Committees of the International Centre for Diarrhoeal Disease Research, Bangladesh. After explaining the purpose of the study, written informed consent was obtained from participants.

# Results

## Characteristics of women

The majority of the respondents (69%) were between 20 and 29 years (Table 1). The mean age of the women was 24.7 years (SD ±4.9). The median year of school attendance was eight for women and their husbands. About one eighth (11.6%) of the respondents had attended secondary school or above. Among the respondents, 33% were in the fourth and fifth wealth quintiles. About 26.7% had four or more antenatal care visits. More than half of the respondents (50.7%) gave birth at home (Table 1).

## Reported illnesses and care-seeking practices for sick neonates

Of the total neonates included in the analysis, 61.2% had suffered from at least one disease. The most common neonatal illnesses reported were fever (36.4%), followed by difficulty in breathing (24.3%), jaundice (19.3%), umbilical cord infection (8.5%), and feeding problem (5.9%) (Fig 2). Among the neonates with any of the seven illnesses, caregivers mostly sought care from village doctors (43%) followed by medical doctors (34%) and homeopaths (17%) (Fig 3). About 35% sought care from skilled service providers, and 61% from unskilled service providers.

**Table 1. Characteristics of study participants.**

| Variables | N = 2,223 |
|---|---|
| | Frequency (%) |
| Mother's age in years | |
| <20 | 275 (12.4) |
| 20–29 | 1537 (69.1) |
| ≥ 30 | 411 (18.5) |
| Mother's education in year | |
| <6 | 520 (23) |
| 6–10 | 1479 (65.4) |
| > 10 | 263 (11.6) |
| Participants' husbands' education in year | |
| <6 | 510 (22.9) |
| 6–10 | 1453 (65.4) |
| > 10 | 260 (11.7) |
| Asset index | |
| One (Poorest) | 470 (21.1) |
| Two | 639 (28.7) |
| Three | 383 (17.2) |
| Four | 295 (13.3) |
| Five (Richest) | 436 (19.6) |
| No. of antenatal care visits | |
| 0–1 | 767 (34.5) |
| 2–3 | 862 (38.8) |
| ≥ 4 | 594(26.7) |
| Place of delivery | |
| Home | 1128 (50.7) |
| Facility | 1095(49.3) |

## Factors associated with skilled healthcare practices for neonatal illnesses

The univariate logistic regression analysis found that the educational level of the participants and their husbands, wealth index, number of antenatal care visits, and place of childbirth were associated with seeking skilled care for neonatal illnesses. All significant factors were included in the

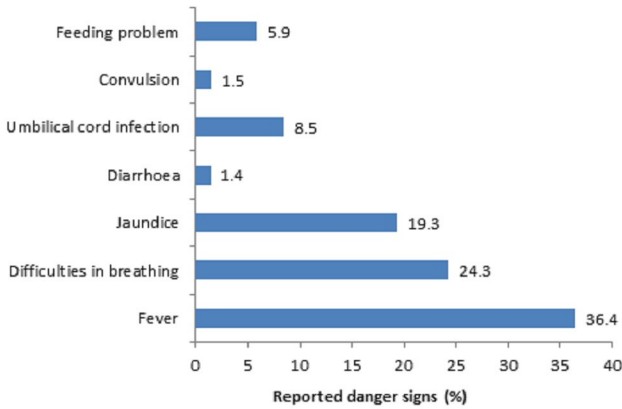

**Fig 2. Neonatal illnesses assessed by reported danger signs experienced within one month of birth.**

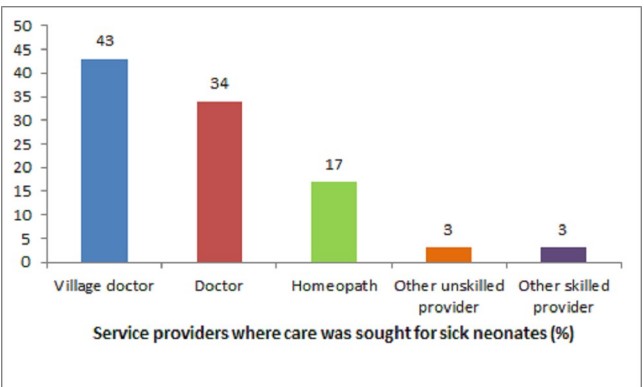

**Fig 3. Types of service providers from whom care was sought for sick neonates.** 'Other unskilled providers' = TBA: Traditional Birth Attendant, Spiritual Healer, Kabiraj, and Family Member. 'Other skilled providers' = Nurse/Midwife, FWV: Family Welfare Visitor, CHCP: Community Health Care Provider, FWA: Family Welfare Assistant, and HA: Health Assistant.

multivariable model. However, the husband's education, number of antenatal care visits, and place of childbirth were found associated with seeking skilled care for neonatal illnesses (Table 2).

In the multivariable logistic regression analysis, the group in which the husbands attended school for more than 10 years had more than two times higher odds (AOR = 2.26, 95% CI: 1.51–3.39) of seeking care from skilled service providers compared to the group with less than 6 years of school attendance. In the group where women attended two or three ANC, and ≥4 ANC, the AOR of seeking skilled care for neonatal illnesses were 1.93 (95% CI: 1.42–2.62) and 2.05 (95% CI: 1.47–2.86), respectively, compared to the women who attended 0 or one ANC. Women who gave birth at a facility were about three folds (AOR = 3.24, 95% CI: 2.50–4.19), more likely to seek skilled care in comparison to those with home delivery.

## Discussion

In this population-based cross-sectional study, we observed that neonates with any reported illness according to the caregivers, fever and breathing difficulties were the most common neonatal danger signs with rates observed about 36% and 24%, respectively. The utilization of skilled healthcare for neonatal sicknesses was low, and only about 35% of mothers contacted with skilled providers for their sick neonates. Furthermore, the study found that women's increased care-seeking from skilled providers for sick neonates was associated with their husbands' higher education level, increased number of ANC visits, and facility delivery.

In the present study, fever was reported to be the most common symptom suffered by neonates, followed by difficulty in breathing. Population-based studies from Bangladesh also reported similar findings of commonly reported symptoms for neonatal illness. The symptoms such as fever and difficulty in breathing might represent respiratory illness such as pneumonia, birth asphyxia, low birth weight/prematurity, and neonatal sepsis, the common causes of both neonatal morbidity and mortality in Bangladesh [2, 24]. Therefore, prompt identification and access to skilled care providers are essential to prevent adverse health outcomes from these reported neonatal illnesses.

Low utilization of skilled healthcare providers was also reported in studies conducted in Bangladesh, wherein less than a quarter of women received skilled care for their sick neonates [2, 12]. A systematic review pointed out the scarcity in data of healthcare seeking patterns for neonatal sickness in LMICs [3]. The care-seeking from trained healthcare providers varied widely in the review and reported the rates between 4% and 100%, with a median of 59% [3].

**Table 2. The associations of selected variables with skilled care utilization for reported neonatal illnesses in Matlab, Bangladesh.**

| Variables | Care-seeking for sick neonates (n = 1,308) | | Crude odds ratio (95% confidence interval) | *Adjusted odds ratio (95% confidence interval) |
|---|---|---|---|---|
| | Skilled care (n = 479) n(%) | Unskilled care (n = 829) n(%) | | |
| Mother's age in years | | | | |
| <20 | 58(12.1) | 97 (11.7) | 1.0 | |
| 20–29 | 346(72.2) | 581 (70.1) | 1.0(0.70–1.42) | |
| ≥ 30 | 75(15.7) | 151 (18.2) | 0.83(0.54–1.27) | |
| Mother's education in year | | | | |
| > 6 | 86(18) | 228(27.5) | 1.0 | |
| 6–10 | 312(65.1) | 551(66.5) | 1.50(1.13–1.99) | |
| > 10 | 81(16.9) | 50(6.0) | 4.29(2.79–6.61) | |
| Participants' husbands' education in year | | | | |
| > 6 | 152(31.7) | 405(48.9) | 1.0 | 1.0 |
| 6–10 | 237(49.5) | 363(43.8) | 1.74(1.36–2.23) | 1.25(0.96–1.63) |
| > 10 years | 90(18.8) | 61(7.4) | 3.93(2.70–5.72) | 2.26(1.51–3.39) |
| Asset index | | | | |
| One (Poorest) | 78(16.3) | 216(26.1) | 1.0 | |
| Two | 109(22.8) | 267(32.2) | 1.13(0.80–1.59) | |
| Three | 95(19.8) | 133(16.0) | 1.98(1.37–2.86) | |
| Four | 81(16.9) | 90(10.9) | 2.49(1.68–3.70) | |
| Five (Richest) | 116(24.2) | 123(14.8) | 2.61(1.82–3.75) | |
| No. of antenatal care visits | | | | |
| 0–1 | 96(20) | 364(43.9) | 1.0 | 1.0 |
| 2–3 | 203(42.4) | 277(33.4) | 2.78(2.08–3.71) | 1.93(1.42–2.62) |
| ≥4 | 180(37.6) | 188(22.7) | 3.63(2.68–4.92) | 2.05(1.47–2.86) |
| Place of delivery | | | | |
| Home | 148(30.9) | 547(66.0) | 1.0 | 1.0 |
| Facility | 331(69.1) | 282(34.0) | 4.34(3.40–5.52) | 3.24(2.50–4.19) |

* Participants' husbands' education, number of antenatal care visits, and delivery place were adjusted to each other in the final model.

The studies with high utilization of skilled care reflected a functional health system with easy accessibility of the health centers in the study areas [13, 27].

A critical aspect of the study findings is that about two-thirds (61%) women sought care from unskilled healthcare providers such as village doctors, homeopaths, spiritual healers, and other non-formal healthcare providers. In Bangladesh, unskilled providers are still frequently chosen based on accessibility, acceptability in the society, lesser costs than skilled care [2]. Furthermore, the perceived belief that the formal health workers are disrespectful to clients with specific characteristics such as the extreme age and non-user of family planning methods may also influence to consult with unskilled healthcare providers [20]. This pattern of choosing unskilled healthcare providers could delay appropriate neonatal illness treatment, resulting in severe morbidities and eventually leading to fatality [20]. The challenge remains how to engage this significant proportion of unskilled healthcare providers into the formal health sectors.

Our study found that educated husbands were two times more likely to seek skilled healthcare for sick neonates compared to less-educated counterparts, which corroborated with other studies [7, 10]. High educational level increases the perception of the severity of neonatal illnesses, strengthens the understanding of health services, and leads prompt decisions about

seeking sick neonates' appropriate care. Furthermore, husbands are more empowered in decision-making, thereby getting easy access to health facilities compared to their uneducated counterparts. The lack of association between women's higher education and seeking skilled care is unexpected, and this is not consistent with studies conducted in Bangladesh [2] and Ethiopia [18]. The observed findings of no association in the present study probably reflect the mothers' limited decision-making role to access skilled healthcare providers.

The numbers of ANC visits were found as a significant predictor for seeking treatment care from skilled service providers in neonatal illnesses. Women who attended four or more ANC visits during their pregnancy were two times more likely to receive care from qualified providers for sick neonates compared to those who attended 0 or one ANC visit. A similar finding was observed in population-based studies conducted in Bangladesh [2, 10]. This positive association explains the ANC visits' role in strengthening caregivers' knowledge and awareness regarding neonatal health and the availability of newborn healthcare services in health facilities [28]. Moreover, women who gave birth at a facility had three times higher odds of care-seeking from trained providers compared to those who gave birth at home, and the finding corroborated with other studies [2, 7, 18]. Giving birth at health institutions probably facilitated postnatal counseling leading to increased awareness of neonatal danger signs and seeking healthcare from skilled providers.

The present study has several strengths. The study was population-based, the participation rate was high, and included relatively large sample size and a population-based cross-sectional design through the study participants' random selection. Also, locally recruited well-trained data collectors collected the data.

There are several limitations to our study. The cross-sectional study design might limit our conclusion on the causal association between predictors and outcome variables. Moreover, the morbidity data were collected based on the women's perception of neonatal illnesses without medical personnel's validation. However, we used easily recognizable common neonatal illnesses, reportable by women themselves. The extended recall period (five to 20 months) might cause recall errors in neonatal illnesses and healthcare seeking practices. However, the errors were probably randomly distributed. Furthermore, the study did not gather information on the quality of care, expenses of health care services, and distance to the health facility, limiting the comprehensive picture of the factors associated with care-seeking from skilled healthcare providers for neonatal sicknesses. This unavailability of possible risk factors may also influence the observed risk estimates in our study. Future studies should consider these social factors to overcome this limitation.

## Conclusion

Our study observed low utilization of care-seeking from skilled healthcare providers for reported neonatal illnesses in the study area. The higher level of husbands' education, increased number of ANC visits, and facility delivery were found as significant predictors for care-seeking from skilled healthcare providers for neonatal illnesses. The policymakers and respective stakeholders should target the husbands with low school attendance for intervention. Furthermore, efforts should continue to increase ANC coverage and facility delivery to improve the coverage of seeking skilled healthcare for neonatal sicknesses and improve neonatal health in Bangladesh and other low-income countries.

## Supporting information

**S1 Dataset.**
(SAV)

## Acknowledgments

We would like to thank all the women for their willingness to participate in the study. icddr,b acknowledges with gratitude the commitment of USAID and PEER Health to its research efforts. icddr,b is also grateful to the Governments of Bangladesh, Canada, Sweden and the UK for providing core/unrestricted support.

## Author Contributions

**Conceptualization:** U. Tin Nu, Jesmin Pervin, Anisur Rahman.

**Data curation:** U. Tin Nu, Jesmin Pervin, A. M. Q. Rahman, Monjur Rahman, Anisur Rahman.

**Formal analysis:** U. Tin Nu, Jesmin Pervin, A. M. Q. Rahman, Monjur Rahman, Anisur Rahman.

**Methodology:** U. Tin Nu, Jesmin Pervin, Anisur Rahman.

**Supervision:** U. Tin Nu, Jesmin Pervin, A. M. Q. Rahman, Monjur Rahman, Anisur Rahman.

**Writing – original draft:** U. Tin Nu, Jesmin Pervin.

**Writing – review & editing:** U. Tin Nu, Jesmin Pervin, A. M. Q. Rahman, Monjur Rahman, Anisur Rahman.

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
