## [Decision Letter · Decision Letter 0]

2 Aug 2020

PONE-D-20-12904

Determinants of Care-seeking Practice for Neonatal Illnesses in Rural Bangladesh: A Community-based Cross-sectional Study

PLOS ONE

Dear Dr. Pervin,

Thank you for submitting your manuscript to PLOS ONE. After careful consideration, we feel that it has merit but does not fully meet PLOS ONE’s publication criteria as it currently stands. Therefore, we invite you to submit a revised version of the manuscript that addresses the points raised during the review process.

We look forward to receiving your revised manuscript.

Kind regards,

Russell Kabir, PhD

Academic Editor

PLOS ONE

Journal Requirements:

2. Please amend either the title on the online submission form (via Edit Submission) or the title in the manuscript so that they are identical.

Reviewers' comments:

Reviewer's Responses to Questions

**Comments to the Author**

1. Is the manuscript technically sound, and do the data support the conclusions?

Reviewer #1: Yes

Reviewer #2: Partly

Reviewer #3: Partly

2. Has the statistical analysis been performed appropriately and rigorously? 

Reviewer #1: Yes

Reviewer #2: N/A

Reviewer #3: No

3. Have the authors made all data underlying the findings in their manuscript fully available?

Reviewer #1: Yes

Reviewer #2: No

Reviewer #3: Yes

4. Is the manuscript presented in an intelligible fashion and written in standard English?

Reviewer #1: Yes

Reviewer #2: No

Reviewer #3: No

5. Review Comments to the Author

Reviewer #1: Line 59-60: How is the situation in Bangladesh should be mentioned

Line 118: Response rate would be important to mention.

Line 125-126: It would be great to know about the number of data collectors, how many data collectors collected the data?

Line 143: Regarding the "covariates" - Are they all adjusted to each other? If so, it should be mentioned in the text and in the table footnotes.

Line 154: In the data analysis section - Statistical software used for the analysis should be mentioned here.

Table 1: Total participant number should be written in the first row of the table.

Table 2: In Variables ≥ 30, Crude odds ratio (95% confidence interval) column shows .83, it should be 0.83

Figure 3: The figure needs to be described in the text (Result section).

Figure 3: Line 197-200: Are these notes of the Figure?

Discussion, Line 201-216: The main findings of the study should be highlighted in this section along with the magnitude of the estimates in percentage.

Discussion, Line 243: Figure 3, It has to be described in the Result section. No need to refer here.

Reviewer #2: This article is interesting but need major revision. Statistical analysis is average. Need to focus on data collection procedure, like sample size selection, study design etc. Need extensive English correction.

Reviewer #3: Thank you for the opportunity to review the manuscript titled " Determinants of Care-seeking Practice for Neonatal Illnesses in Rural Bangladesh: A Community-based Cross-sectional Study". This manuscript examines the factors associated with seeking care from skilled healthcare service providers for reported neonatal illnesses in rural Matlab, Bangladesh

Overall, this manuscript can contribute to readers understanding on Care-seeking Practice for Neonatal Illnesses and research gap in the literature. However, this manuscript lacks of important components in methodology, analysis' description and discussion. My review of the manuscript and recommendations to the authors are enclosed.

6. PLOS authors have the option to publish the peer review history of their article (what does this mean?). If published, this will include your full peer review and any attached files.

Reviewer #1: No

Reviewer #2: No

Reviewer #3: **Yes: **Md. Golam Dostogir Dostogir Harun

---

## [Author Response · Author response to Decision Letter 0]

15 Sep 2020

Manuscript title: Determinants of Care-seeking Practice for Neonatal Illnesses in Rural Bangladesh: A Community-based Cross-sectional Study

Manuscript ID: PONE-D-20-12904

Thank you for giving us the opportunity to submit the revised version of our manuscript. Please see below the reviewer’s comments followed by our responses. 

Response to Reviewer's comments:

Reviewer 1: 

Comments:

Line 59-60: How is the situation in Bangladesh should be mentioned

Response: Thank you for your advice. We have added additional texts (page 4, line 64-65).

Line 118: Response rate would be important to mention.

Response: We have added the response rate (page 7, line 141). 

Line 125-126: It would be great to know about the number of data collectors, how many data collectors collected the data?

Response: We have added information on the number of data collectors (page 8, line 147-148).

Line 143: Regarding the "covariates" - Are they all adjusted to each other? If so, it should be mentioned in the text and in the table footnotes.

Response: Thanks for the comments. The factors were adjusted to each other. We have added the information in the methods (page 9-10, line 181-186) and also in the table footnote (page 15, line 224-225).

Line 154: In the data analysis section - Statistical software used for the analysis should be mentioned here.

Response: We have added this (page 10, line 188-189).

Table 1: Total participant number should be written in the first row of the table.

Response: We have added accordingly (page 11, Table 1).

Table 2: In Variables ≥ 30, Crude odds ratio (95% confidence interval) column shows .83, it should be 0.83

Response: We have revised accordingly.

Figure 3: The figure needs to be described in the text (Result section).

Response: Thanks for the observation. We have added texts related with the figure (page 12-13, line 209-210).

Figure 3: Line 197-200: Are these notes of the Figure?

Response: These are figure legends that we have added according to the journal guideline. 

Discussion, Line 201-216: The main findings of the study should be highlighted in this section along with the magnitude of the estimates in percentage.

Response: We have revised the paragraph and presented the main findings from the study (page 16, line 240-246).

Discussion, Line 243: Figure 3, It has to be described in the Result section. No need to refer here.

Response: We have described the figure in the result section (page 12-13, line 209-210). Further, we have revised the text in the discussion section (page 17, line 265-267).

Reviewer 2:

Comments

1. In Background of Abstract, how poor utilization of skilled care services in neonatal illnesses is crucial to 18 reduce neonatal morbidity and mortality?

Response: We have corrected this (Page 2, Line: 17-18).

2. In second para of Introduction, "Available literature has reported that effective utilization of skilled health care services for 67 newborns remains low" where the research conducted?

Response: We have revised the texts and added the study area (Page 4, Line 74-75). 

3. In the same para "although the proportion of home-based delivery is reduced significantly" what does it mean kind of poor structure of the sentence.

Response: We have revised the text (Page 4, Line 70-72)

4. Poor level of English and inconsistencies appeared in introduction section.

Response: We have revised the text and checked the language carefully. Further, we used the professional Editage service to make the language clear and free of errors in the revised version. 

5. What type of individual and social factors are responsible for the utilization of a skilled health care provider?

Response: We have added the texts accordingly (Page: 5, line: 80-86).

6. This study did not provide any previous evidence however some literatures addressed this issue in Bangladesh and other countries. 

Response: We have added texts (Page 4-5, Line 74-76).

7. Study gap is not clear including research question and theoretical framework.

Response: We have added new text to address the gaps more clearly (page 5-6, line 91-100). We have not added the theoretical framework as we have discussed at length how the available factors may influence in uptake of care.

8. In methods, sample size selection is not clear, how the authors selected sample, like, any formula, exclusion and inclusion criteria etc.

Response: We have revised the text to make the sample selection clear (Page 7-8, Line 132-143). 

9. In discussion, line 245, the influential role of cultural beliefs and social stigma… what types of cultural beliefs and social stigma…

Response: We have revised the text to make this clear (Page 17, Line 268-271).

10. In line 246, "severe consequences" what are they

Response: We have revised the text (Page 17-18, Line 271-273).

11. What are the new policies add to this study.

Response: We have added the texts (page 20, line 322-326). 

12. What is the difference between this study and “Chowdhury SK, Billah SM, Arifeen SE, Hoque DME (2018) Care-seeking practices for sick neonates: Findings from cross-sectional survey in 14 rural sub-districts of Bangladesh. PLoS ONE 13(9): e0204902”…

Response: Thanks for the comments. The above article was based on data collected after the period of an intervention. Therefore, probably not represent the baseline situation of care seeking pattern and its determinants in the study area. We have now mentioned this in the text (page 5, line 96-98). 

13. Overall cross check the table and text values and need comprehensive English language correction.

Response: Thanks for the suggestion. We have carefully checked and revised accordingly. Further, we used the professional Editage service to make the language clear and free of errors in the revised version.

Reviewer 3: 

Comments:

Abstract

Line 19, 27: Comment: Objectives and findings of the study are matched with a study conducted in rural Bangladesh in 2001 by Ahmed S et al. 

Reference:Ahmed S, Sobhan F, Islam A. Neonatal morbidity and care‐seeking behavior in rural Bangladesh. Journal of tropical pediatrics. 2001;47(2): 98-105

Response: Thanks for your observation. The study conducted by Ahmed et al. (2001) had a similar objective as our study, but this study was conducted during the period January 1996–August 1998 which is 15 years earlier than the present study. Therefore, the study does not truly reflect the current situation on this important public health issue in the country. We have also added new text to address this (page 5, line 96-98).

Line 23: Comment: Please mention the name of the method that helps to find the adjusted odds ratio.

Response: We have added the method in abstract (page 2, line 23-25). However, we have added new text in the methods to make this clear (page 9-10, line 181-186).

Line 23; the sentence should be rephrased 

Response: We have revised the text. (Page 2, Line 23-25).

Introduction; The introduction part has space to improve significantly with more relevant and recent references. The is also gap in consistence of the writing flow.

Response: We have revised the introduction section and added recent references.

Materials and methods

Line:117: Comment: Please discuss the technique of selecting 2483 women randomly from 6,741 women. Without any explanation of a specific reason, there will be the possibility of selection bias. 

Response: We have added new text to make this clear (Page 7- 8, line 132-143).

Line 143: Comment: Add PNC visit as a covariate because it has an impact on mother knowledge about neonates illness. See the reference. 

Reference : Bulto GA, Fekene DB, Moti BE, Demissie GA, Daka KB. Knowledge of neonatal danger signs, care seeking practice and associated factors among postpartum mothers at public health facilities in ambo town, Central Ethiopia. BMC research notes. 2019;12(1):549.

Response: We agree with the comments, and mentioned PNC in the introduction section (page 5, line 81-83). Further, we have mentioned the role of post-natal counseling of healthcare seeking patters (page 19, line 296-298). However, in this study area after excluding the facility delivery rate less than 4% used PNC, therefore limiting the possibility to include PNC in the the analysis.

Line 159: Comment: Please mention which model selection technique was followed to select determinants. The method that was used has the problem of overestimation. It didn’t say the way of optimizing the overestimation problem. 

Response: We have revised the texts (page 9-10, line 181-189). Although odds ratio may inflate the effect estimates, however use of logistic regression is a common methods for risk estimation. Further, the exclusion of economic status (by asset index) may influence the observed ORs. As said in the methods, asset index was not significant and thus excluded from the final model. However, the risk estimates might be high due to the unavailability of a number of social and behavioral factors. We have mentioned this weakness in the discussion section (page 19, line 314-315).

Line 160: Comment: Please mention the covariates in the method section or table that was used in the final model to get adjusted odds ratios.

Response: We have addressed this clearly in the methods section (page 9-10, line 181-189) and also by a footnote on table 2 (page 16, line 224-225).

Result:

Line 172, 173 Comment: Mean age and median year of schooling were not found in table 1. Please check is it the number of school attendance or year of school attendance? 

Response: We have revised the text so it is clear that the means/median is not reflecting the table (page 10, line 198-199). However, it is not uncommon to add additional information as text.

Line 180: Comment: Repetition

Response: We have revised the texts (page 12, line 206). 

Discussion: 

Discussion is too long, please try to reduce the texts. Please use recent reference from similar settings. Need to improve the language as well. 

Response: We revised the discussion section and revised the references. Also, we used the professional Editage service to improve the language and free of errors.

Line 214, 215, 216 

Comment: Repetition of result

Response: We have revised the texts (page 16, line 244-246).

Line 267

Comment: Not only ANC but also PNC helps to strengthen mother knowledge about danger sign of neonate. See the below reference

Referance : Bulto GA, Fekene DB, Moti BE, Demissie GA, Daka KB. Knowledge of neonatal danger signs, care seeking practice and associated factors among postpartum mothers at public health facilities in ambo town, Central Ethiopia. BMC research notes. 2019;12(1):549.

Response: As already mentoned above, we agree with the comments, and added PNC in the introduction section (page 5, line 81-83). Further, we have mentioned the role of post-natal counseling of healthcare seeking patters (page 19, line 296-298). However, in this study area after excluding the facility delivery rate less than 4% used PNC, therefore limiting the possibility to include PNC in the the analysis.

Better to include Strength and Limitation of the study

Response: We have already mentioned the strength and limitation of the study. However we revised the texts (page 19, line 300-316).

Conclusion: Conclusion should be based on study findings. 

Response: We have revised the texts (page 20, line 319-326).

---

## [Editor Report · Decision Letter 1]

24 Sep 2020

Determinants of Care-seeking Practice for Neonatal Illnesses in Rural Bangladesh: A Community-based Cross-sectional Study

PONE-D-20-12904R1

Dear Dr. Pervin,

We’re pleased to inform you that your manuscript has been judged scientifically suitable for publication and will be formally accepted for publication once it meets all outstanding technical requirements.

Kind regards,

Russell Kabir, PhD

Academic Editor

PLOS ONE
---

## [Editor Report · Acceptance letter]

30 Sep 2020

PONE-D-20-12904R1 

*Determinants of Care-seeking Practice for Neonatal Illnesses in Rural Bangladesh: A Community-based Cross-sectional Study*

Dear Dr. Pervin:

I'm pleased to inform you that your manuscript has been deemed suitable for publication in PLOS ONE. Congratulations! Your manuscript is now with our production department. 

Kind regards, 

on behalf of

Dr. Russell Kabir 

Academic Editor

PLOS ONE